# Diversity and Exploration of Endophytic Bacilli for the Management of Head Scab (*Fusarium graminearum*) of Wheat

**DOI:** 10.3390/pathogens11101088

**Published:** 2022-09-23

**Authors:** Noyonika Kaul, Prem Lal Kashyap, Sudheer Kumar, Deepti Singh, Gyanendra Pratap Singh

**Affiliations:** 1Division of Crop Protection, ICAR-Indian Institute of Wheat and Barley Research, Karnal 132001, Haryana, India; 2Amity Institute of Microbial Technologies, Amity University, Karnal 132001, Haryana, India; 3ICAR-Indian Institute of Wheat and Barley Research, Karnal 132001, Haryana, India

**Keywords:** *Bacillus*, biological control, disease index, endophyte, wheat head scab

## Abstract

*Fusarium graminearum* causing head scab (HS) or head blight (HB) disease in wheat is one of the nasty fungi reported to cause significant grain quality and yield loss. Biological control using endophytic bacteria has emerged as a prospective option for containing fungal diseases in an environmentally benevolent, durable, and sustainable manner. In this regard, 112 endophytic bacilli were isolated from the anthesis stage (Zadok’s growth stage 65) from five different wheat genotypes with an aim to identify prospective antagonistic strains against *F. graminearum*. The molecular identity of the strains was confirmed by matching 16S rRNA sequences of bacterial strains with the gene sequences of type strains available in the National Center for Biotechnology Information database and reported 38 different species of *Bacillus* in all the five wheat cultivars. Further, it has been observed that only fourteen strains (*B. clarus* NOK09, *B. mojavensis* NOK16, *B. subtilis* NOK33, *B. rugosus* NOK47, *B. mojavensis* NOK52, *B. clarus* NOK59, *B. coahuilensis* NOK72, *B. cabrialesii* NOK78, *B. cabrialesii* NOK82, *B. rugosus* NOK85, *B. amyloliquefaciens* NOK89, *B. australimaris* NOK95, *B. pumilus* NOK103, and *B. amyloliquefaciens* NOK109) displayed in-vitro antagonistic effect against *Fusarium graminearum* fungus. Furthermore, the three endophytic *Bacillus* strains showing the strongest antagonistic effect (>70% of growth inhibition of fungal mycelium) under in-vitro antagonistic assay were selected for field experiments. In a two-year consecutive field study, a combination of three strains (*B. clarus* NOK09 + *B. subtilis* NOK33 + *B. amyloliquefaciens* NOK109) displayed a remarkable reduction in HS disease index by 81.47% and 77.85%, respectively. Polymerase chain reaction assay detected three genes (*itu*D, *bmy*C, and *srf*A) involved in antibiotic biosynthesis pathways. Additional attributes such as potassium solubilization, siderophore release, and hydrolytic enzyme (protease, lipase, amylase, chitinase, and pectinase) synthesis have been observed in these strains. Overall, the present study was successful in profiling endophytic bacilli and selecting the combination of effective antagonistic endophytic *Bacillus* strains that could be the best alternative for the sustainable and ecological sound management of HS disease in wheat under field conditions.

## 1. Introduction

*Fusarium**graminearum* causing head scab (HS) or head blight (HB) disease in wheat is one of the nasty fungi reported to cause significant grain quality and yield loss. The major outbreaks of HS disease in wheat from major wheat-growing countries of the world, including Australia, Asia, Canada, Europe, and South America, have been well documented [1]. In India, noteworthy wheat yield loss has been noticed due to HS disease, specifically when rain coincides with the anthesis stage of the wheat [2]. In wheat fields, HS symptoms can be seen on the glumes and rachis as water-soaked spots. Later, the disease dispersed within the wheat ear heads, and infected ear heads showed partial to complete bleaching. Under warm, humid conditions, orange sporulation can also be seen on infected wheat spikelets. Published literature revealed that crop rotation, agronomic interventions, and avoidance of the cultivation of susceptible varieties are not useful in providing an adequate level of protection against diseases. Moreover, none of these approaches alone or in an integrated manner can control the disease in an effective manner [3,4,5]. Growing HS tolerant cultivars is one of the most effective options. However, in India, a majority of the varieties under cultivation have a low level of resistance against *F. graminearum* due to a lack of complete HS-resistant wheat cultivars. Although adequate levels of wheat protection from HS can be achieved with agrochemicals and resistance inducers, their efficacy varies with each fungal strain [6,7,8,9,10]. Moreover, recommended fungicides had adverse effects on soil microbiota and the environment, warranting new technological innovations with minimal reliance on agrochemicals [11]. Recently published reports indicated that agrochemicals are regularly losing their ground in the management of diseases under field conditions due to the rapid evolution of pathogen races and mutations resulting in resistance phenotypes [12,13] and thus warrants alternative molecules with novel modes of action. Recently, de Chaves et al. [14] noticed Tebuconazole and Prochloraz resistant isolates of *F. graminearum* in wheat fields, where the continuous application of fungicides was utilized for a long time. At present, a limited variety of novel molecules for tackling the rising problem of fungicide resistance development in *F. graminearum* is available. Moreover, the successful and timely management of HS by fungicides is difficult to attain as the effect of fungicide is highly influenced by the optimal timing of the fungicide application, which is too short in the case of HS disease [3]. Therefore, HS management by employing eco-compatible technologies such as antagonistic endophytic bacteria could be of great benefit.

Several studies highlighted the escalating research interest in exploring the prospective role of endophytic bacteria in mitigating the adverse effects on agricultural crops as a consequence of biotic and abiotic stressors [15,16]. Generally, endophytic bacteria reside inside the plant host without displaying any prominent disease symptoms. Here, it is important to mention that the prime benefit of harnessing endophytes as potential antagonists is their strong adaptation character to stay within the plants, which makes them suitable candidates as crop stress defenders from biotic anomalies [17]. The majority of the benefits of endophytic bacteria are analogous to rhizospheric bacteria. However, emerging research evidence ranked endophytic bacteria over traditional antagonists due to their ability to transfer to the next generation in sustainable manner [18,19]. Among various endophytic bacteria, the exploration of endophytic antagonistic *Bacillus* has emerged as one of the highly prospective and eco-friendly substitutes due to its unique inherent endospore formation capacity. This capacity helps to get high resistance against stressors, omnipresence, and stability in extreme environments [20,21]. Besides this, numerous studies illustrate the remarkable contribution of antimicrobial peptides (AMPs) released by *Bacillus* strains in the microbial management of fungal plant pathogens [22,23,24,25,26]. Unfortunately, inadequate information exists with respect to the occurrence of genes associated with the synthesis of AMPs in *Bacillus* species allied with wheat hosts as endophytes.

There have been a number of reports of functional characterization and evaluation of endophytic *Bacillus* strains for the management of fungal pathogens attacking different crops. For instance, endophytic *Bacillus mojavensis* has been reported to enhance maize growth when attacked with *Fusarium verticillioides* [27]. Similarly, Pan et al. [17] noticed that endophytic *B. megaterium* and *B. subtilis* derived from wheat grain markedly suppressed the fungal growth of *F. graminearum*. Besides these, several other *Bacillus* strains of endopphytic origin, such as *B. amyloliquefaciens* YN201732 [28], *B. velezensis* strain OEE1 [29], *B. thuringiensis* [30], and *B. safensis* B21 [31] have been recognized as ideal candidates for the bio-control of myriads of fungal pathogens attacking diverse types of agricultural crops. In a recent study, Munakata et al. [32] made a functional comparison of endophytic microbiota of vetiver root from different ecological niches across Africa and Europe and noticed a strong antagonistic character of *Bacillus* origin endophytes towards *F. graminearum*. All these research efforts indicate the wide spectrum bio-control potential of endophytic *Bacillus* strains. However, despite the huge economic significance of wheat, very limited research efforts have been made to explore the diversity of endophytic *Bacillus* strains as potential antagonists for the management of wheat diseases in an ecologically sound manner. In wheat, the anthesis period is identified as the most susceptible stage for *F. graminearum* infection. Further, it has been observed that anthers serve as a frequent path of entrance into the wheat host and markedly important growth stage for the application of bio-control agents or fungicides to restrict the fungal infection in a spatio-temporal manner, even under favorable environmental conditions [33]. Thus, the present study was planned with following prime objectives: (i) to profile the endophytic bacilli associated with ears of different wheat genotypes and evaluate their antagonistic features towards *F. graminearum,* and (ii) to assess the potential of identified antagonistic endophytic *Bacillus* species individually or in combination for the bio-control of HS under field conditions.

## 2. Materials and Methods

### 2.1. Field Sampling for the Isolation and Identification of Endophytic Bacillus

Healthy ear head samples were collected from wheat genotypes (*viz*., DBW187, HD2967, PBW343, HD3086, and Agra local). Every sample comprised of three plants of 65 days old wheat crop (anthesis stage) was gathered in sterilized plastic sampling bags. A complete surface sterilization procedure reported by Kushwaha et al. [15] was employed for the isolation of endophytic bacteria from wheat ear heads. In brief, preliminary treatment of sampled wheat ear heads was performed with tap water, followedwashing the sample with double distilled sterilized water. After preliminary washing, treated samples were placed in 70% ethanol for 1 min and consequently soaked in 3% sodium hypochlorite for 3 min and 70% ethanol again for 30 s in a sequential manner. The final treatment of the sample was done with autoclaved double distilled water. This step was repeated three times before the final drying of the treated samples on sterile Whatman filter paper. Approximately 1 g of each tissue was finely crushed and suspended in phosphate-buffered saline (PBS) solution (Hi-Media Laboratories Pvt Ltd. India). Heat treatment of the macerated tissue suspension was performed for the isolation of *Bacillus* species, according to Sharma et al. [34]. An aliquot (100 mL) from each heat-treated serial dilution (10^−2^ to 10^−6^) sample was streaked on Petri plates containing different media [e.g., nutrient agar (NA), Luria Bertani agar, and tryptic soy agar] and incubated at 37 ± 2 °C temperature for two days. An aliquot (100 µL) of remaining water after the last wash was inoculated on NA Petri plates to inspect the occurrence of different bacterial colonies for seven days. Different microscopy and biochemical tests (Gram staining, motility, microscopic appearance, oxidase test, catalase test, reduction of nitrate to nitrite) were also conducted in three replicates for preliminary confirmation of the *Bacillus* spp. [35]. Bacterial colonies with different morphotypes were selected and maintained in glycerol (50%) at −80 °C [36].

For the accurate identification of bacterial endophytes and detection of genes linked with antimicrobial peptides (AMPs) produced by *Bacillus*, isolation of the total genomic DNA of each bacterium was carried out as per the protocol of Pospiech and Neumann [37]. Amplification of 16S rRNA region and AMP genes was carried out in a thermocycler machine (Q cycler 96, Hain Lifescience UK Ltd., Surrey, UK), and conditions of PCR reaction are described in Table 1. For the identification of bacterial endophytes, generated amplicons of ~1500 bp were sent to Eurofins genomics sequencing services, India, for DNA sequence analysis. The matching of 16S rRNA gene sequences of bacterial endophytes was made against the sequences available in the EZ Biocloud e-server (https://www.ezbiocloud.net/, accessed on 10 December 2021) to identify their nearest match. Molecular Evolutionary Genetics Analysis (MEGA) version 11 [38] tool was used for the construction of phylogenetic tree with nearest type strain sequences of *Bacillus* sp. available in National Center for Biotechnology Information (NCBI; https://www.ncbi.nlm.nih.gov/, accessed on 2 February 2022) database. The gene sequences were deposited in the NCBI databank to get the gene accession numbers (Figure 1, Figure 2, Figure 3, Figure 4 and Figure 5). The alignments of the 16S rRNA gene sequences of bacterial endophytes, along with matching type strains, were conducted with the help of Clustal W [39]. The multiple sequence alignment profiles were used to build the best fit phylogenetic tree using the neighbor-joining method [40] with Kimura’s two-parameter model [41] executed in MEGA version 11 software [38]. Bootstrap analysis was performed to assess confidence levels for the branches with 1000 replicates [42].

### 2.2. In-Vitro Determination of Antagonistic Activity 

Dual-culture plate assay was used to confirm the antagonistic capabilities of endophytic bacterial strains against *F. graminearum*. Briefly, a 5 mm diameter segment of fungal mass of highly virulent *F. graminearum* NFG1 isolate was positioned in the mid-point of Petri plates amended with PDA: NA (1:1). Each strain (~5 × 10^8^ cfu ml^−1^) was streaked in a straight line closer to the border of Petri plate. The inoculated plates were incubated at 30 ± 2 °C. *F. graminearum* inoculated Petri plate without endophytic strain, and neutral bacterial strain (*Bacillus pumilus* NOK68) served as a control. Data on the growth of *F. graminearum* was recorded in mm till the control plate (without endophytic strain) was completely filled. *F. graminearum* growth inhibition (%) by each endophytic strain was computed by the formula quoted by Sharma et al. [34]. The assay was performed with three independent repetitions.

### 2.3. Field Trial of Antagonists against FHB 

Three *Bacillus* strains that reflected maximum antagonistic action towards *F. graminearum* under in-vitro conditions were chosen for testing their bio-control potentialities against HS disease in susceptible wheat genotype (PBW 343) at Crop Protection Experiment Area, ICAR-Indian Institute of Wheat and Barley Research, Karnal, Haryana, India. Under each field treatment, wheat seeds were sown in six lines (each line of 3 m long) with 22.5 cm line spacing. All the experiments were planned in a randomized complete block design with five replicates per treatment. Different treatments comprised of antagonistic endophytic bacteria (~10^7^ cfu ml^−1^) and without endophytic bacteria were used to inoculate 150 wheat ear heads (50 heads per replicate, three replicates per treatment) at anthesis time (Zadoks growth stage 65). The bacterial suspension (2 mL per wheat ear head) in each plot was applied by a compressed air sprayer in the late afternoon (at 4:00PM). The control treatment (T8) was only sprayed with highly virulent *F. graminearum* isolate (NFG1). An additional control treatment (T9) comprised of fungicide (Propiconazole @ 0.1%, Syngenta, Pune, India) sprayed alone. Field assessment of HS incidence and HS severity was made at the late milk stage. Disease data was recorded on 40 ear heads per replicate (200 ear heads treatment ^−1^) as per the disease rating described by Stack and McMullen [46]. Disease index (DI) computation was made by using the formula: (incidence × severity)/100).

### 2.4. Strain Characterization for Siderophore Production and Hydrolytic Enzyme Activities

Chrome azurol (CAS) agar assay [47] was carried out to know the potential of 14 endophytic antagonistic bacterial strains for siderophore synthesis. For assessing the potential of endophytic bacterial strains for potassium (K) solubilization, the spot inoculation procedure quoted by Hu et al. [48] was used, where potassium feldspar powder was used as an insoluble phosphate source. The potential of endophytic bacterial strains for siderophore production and potassium solubilization was checked for halo development around the colony, and halo size was recorded in mm. Similarly, the potential of the extracellular hydrolytic enzymes (chitinase, amylase, cellulase, protease, and lipase) production by each strain was determined according to Kushwaha et al. [21] and Sharma et al. [34], where the formation of clear halo zone by each strains was observed on NA Petri plates amended with a substrate such as chitin, soluble starch, carboxy methyl cellulose, casein, and tributerin for the detection of chitinase, amylase, cellulase, protease, and lipase activities, respectively. The halo zone around the bacterial colony was indicative of positive enzymatic activity and measured in mm. Each assay was repeated three times.

### 2.5. Detection of Antimicrobial Peptide Gene (s)

Three endophytic strains (NOK9, NOK33, and NOK109) displaying strong antifungal activities (>70%) were tested for the presence of different AMP genes in *Bacillus* strains. The polymerase chain reaction temperature profiles and primers information given in Table 1 was used for the amplification of the surfactin, bacillomycin, and iturin genes, respectively. 

### 2.6. Statistical Analysis

The field experiments performed to check the antagonistic effects of selected bacteria were arranged in complete randomized block design (CRD) with three replicates. A one-way analysis of variance (ANOVA) was performed to test the significance of each treatment. A post hoc comparison of mean values was carried out by performing Duncan’s multiple range test (DMRT).

## 3. Results

### 3.1. Species Diversity of Endophytic Bacillus in Tissues of Wheat of Different Genotypes

The 16S rRNA amplicon sequencing results illustrated that all the 112 strains isolated from wheat ear heads of five different cultivars (*viz*., DBW187, HD2967, PBW343, HD3086, and Agra local) represent the *Bacillus* genus. The genotypes DBW187, HD2967, PBW343, HD3086, and Agra local contain 24, 27, 17, 26, and 18 *Bacillus* strains of endophytic origin, respectively. The bacterial strains belonged to 28 different species (*B. aerophilus*, *B. albus*, *B. altitudinis*, *B. amyloliquefaciens*, *B. atrophaeus*, *B. australimaris*, *B. badius*, *B. cabrialesii*, *B. cereus*, *B. clarus*, *B. coahuilensis*, *B. dafuensis*, *B. ferrooxidans*, *B. fungorum*, *B. glycinifermentans*, *B. haikouensis*, *B. halotolerans*, *B. haynesii*, *B. licheniformis*, *B. mojavensis*, *B. mycoides*, *B. nakamurai*, *B. paralicheniformis*, *B. paramycoides*, *B. pseudomycoides*, *B. pumilus*, *B. rugosus*, *B. safensis*, *B. siamensis*, *B. stratosphericus*, *B. subtilis*, *B. subtilis* subsp. *spizizenii*, *B. swezeyi*, *B. tequilensis*, *B. velezensis*, *B. wiedmannii*, *B. yapensis*, *B. zanthoxyli*, and *B. zhangzhouensis*) and reveal a high degree (100%) of sequence resemblance with the type strain sequences in available in EZ Biocloud e-server (https://www.ezbiocloud.net/, accessed on 10 December 2021). It has been observed that the wheat genotype DWB 187 contained the maximum number of distinct *Bacillus* species (20 species; Figure 1) followed by HD 2967 (19 species; Figure 2), DBW 343 (12 species; Figure 3), HD 3086 (17 species; Figure 4), and Agra local (13 species; Figure 5). Additionally, the gene sequences of 16S rRNA were deposited at NCBI GenBank with accession number mentioned in Figure 1, Figure 2, Figure 3, Figure 4 and Figure 5.

### 3.2. Identification and Selection of Antagonistic Endophytes

A total of 112 endophytic *Bacillus* strains were recovered from five different wheat cultivars, but only fourteen strains (*B. clarus* NOK09, *B. mojavensis* NOK16, *B. subtilis* NOK33, *B. rugosus* NOK47, *B. mojavensis* NOK52, *B. clarus* NOK59, *B. coahuilensis* NOK72, *Bacillus cabrialesii* NOK78, *B. cabrialesii* NOK82, *B. rugosus* NOK85, *B. amyloliquefaciens* NOK89, *B. australimaris* NOK95, *B. pumilus* NOK103, and *B. amyloliquefaciens* NOK109) displayed antagonist action towards highly virulent *F. graminearum* NFG1 isolate (Table 2). The strain NOK68 did not show any antagonistic effects against *F. graminearum* NFG1 (Figure 6). All the antagonists presented a significant reduction in the *F. graminearum* growth (*p* < 0. 05), among which *B. clarus* NOK09 (77.3% *F. graminearum* growth inhibition), *B. subtilis* NOK33 (71.9% *F. graminearum* growth inhibition), and *B. amyloliquefaciens* NOK109 (79.4% *F. graminearum* growth inhibition) displayed more than a 70 % inhibitory effect on fungal growth (Table 2). 

### 3.3. Strain Characterization for Potassium Solubilization, Siderophores Release and Hydrolytic Enzyme Activity

Among fourteen shortlisted strains on the criterion of in vitro antifungal activity, NOK09, NOK16, NOK33, NOK47, NOK52, NOK59, NOK72, NOK78, NOK82, NOK85, NOK89, NOK95, NOK103, and NOK109 were found siderophore producing in nature and displayed clear halo zone (10.11–21.11 mm) formation (Table 2). It has been noticed that strain NOK109, followed by NOK09 and NOK33, released an amazingly high amount of siderophore and displayed more than 20.62 mm clear halo zone formation. Likewise, NOK09, NOK16, NOK33, NOK47, NOK52, NOK59, NOK78, NOK82, NOK89, NOK103, and NOK109 strains were observed as efficient phosphate solubilizers, showing a prominent, clear halo zone in the range of 9.42 mm to 17.86 mm. Similarly, NOK09, NOK33, NOK47, NOK52, NOK59, NOK78, NOK85, NOK89, NOK95, NOK103, and NOK109 were recorded as lipase producers, displayinga halo zone in the range of 10.71 mm to 22.07 mm. Besides this, all the fourteen strains had the potential to synthesize extracellular enzymes (protease, amylase, chitinase, and pectinase) at variable levels (Table 2). Three strains, i.e., NOK9, NOK33, and NOK109, showed the highest activities for protease, lipase, amylase, chitinase, and pectinase (Table 2).

### 3.4. Detection of Antimicrobial Peptides (AMPs) Biosynthesis Associated Genes

PCR amplification of AMP genes in three endophytic *Bacillus* strains (*B. clarus* NOK09, *B. subtilis* NOK33, and *B. amyloliquefaciens* NOK109) were displayed in Figure 7. Amplification of the iturin (*itu*D), bacillomycin (*bmy*C), and surfactin (*srf*A) gene produced a single specific amplicon of 647, 395, and 201 bp, respectively, in NOK9, NOK33, and NOK109 strains.

### 3.5. Field Evaluation of Antagonistic Strains against F. graminearum

Three endophytic *Bacillus* strains (*B. clarus* NOK09, *B. subtilis* NOK33, and *B. amyloliquefaciens* NOK109) which showed the greatest bio-control activity (>70%) towards *F. graminearum* NFG1 under in-vitro conditions were chosen to appraise their bio-control potential to contain HS disease in wheat under field conditions and observations were presented in Figure 8, Figure 9 and Figure 10 and Table 3. The perusal of data indicated that in 2020–2021, the disease severity varied between 15.29 to 87.92 % of infected spikelets (Figure 9), while in 2021–2022, it ranged from 15.05 to 87.80% (Figure 9). The wheat plants sprayed at the anthesis stage with different treatments of endophytic antagonists individually (NOK09, NOK33, and NOK109) or in combination (NOK09 + NOK33, NOK33+. NOK109, NOK09 + NOK109, and NOK09 + NOK33+ NOK109) significantly reduced the impact of HS incidence, severity, and disease index compared to the control check i.e., T8 (sprayed with only highly virulent *F. graminearum* NFG1 isolate) in both the years (Figure 8 and Figure 9). A noteworthy decline in DS was recorded in plots sprayed with a combination of three strains composed of NOK 9 + NOK 33 + NOK 109 (61.18 and 59.14%) followed by NOK 9 + NOK109 (50.96 and 50.12%), NOK 33 and NOK 109 (45.57 and 45.80%), NON 9 + NOK33 (42.72 and 41.63%), NOK 109 (40.02 and 35.64%), NOK 9 (27.32 and 24.42%) and NOK33 (33.20 and 30.05%), in both the years, respectively. Similar trends were observed in the case of disease incidence in both years (Figure 8). It has been noticed that among the treatment of individual strains, a significant reduction in DI (Figure 8), DS (Figure 9), and FHB index (Table 3) was observed in the case of NOK109 followed by NOK33 and NOK9 relative to untreated control. Based on field data from both years, all the treatments of antagonists (T1 to T7), either singly or in combination, show a significant gain in grain yield (*p* < 0.05) relative to the untreated endophyte control (T8) (Figure 10). Maximum disease reduction effects were observed for T9 (Propiconazole @ 0.1%), which reduced disease severity by 82.62 and 82.90% in both years, respectively (Table 3). In both the years, treatment with NOK09 + NOK33 + NOK109 was the most effective among all antagonist treatments.

## 4. Discussion

Head scab (HS) is acknowledged as one of the prime diseases of wheat because of its global presence in all wheat-growing countries and is ranked as the fourth biggest threat to successful quality wheat production [49]. Due to the unavailability of completely resistant cultivars, the management of HS is heavily relying on fungicides. Nevertheless, with injudicious and excessive use of recommended fungicides, the cases of resistance development in *F. graminearum* against fungicides are mounting and ultimately resulted in the loss of their field efficacy [6,50]. Furthermore, the farmers many times miss the optimal fungicide application time due to their inability to predict the right infection time of the wheat spike. As a result, it becomes necessary to devise new strategies for the effective and timely management of HS disease in wheat. Therefore, the major goal of the present research is to decipher the diversity and antagonistic potential of endophytic *Bacillus* spp. allied to the anthesis stage of wheat for the eco-friendly and sustainable management of HS under natural field conditions. In the current research investigation, the diversity of cultivable endophytic *Bacillus* strains allied with healthy wheat ear heads of five different wheat cultivars (DBW187, HD2967, PBW343, HD3086, and Agra local) has been explored. It is important to mention here that healthy wheat ears were selected for the exploration of *Bacillus* endophytes because of the fact that the healthy plant parts harbor a diverse and distinct type of microbial population of endophytic bacteria than unhealthy plants [51,52]. Earlier published literature reported that the composition of bacterial endophytes was greatly influenced by plant type, growth stage, and soil nutrient availability [53]. Marag et al. [54] reported the flowering stage as the prime site for maximal abundance and population of endophytic bacteria than other crop growth stages. Most importantly, endophytic *Bacillus* strains displaying excellent antagonistic activity in different crops have been reported by various workers [15,55,56,57]. It is important to mention that the anthesis or wheat ear emergence stage is the most susceptible window for *F. graminearum* infection [33]. Moreover, because phyllosphere endophytes reside in a similar ecological niche as foliar pathogens [58], there is a great probability that the bacterial endophytes associated with wheat ear head might serve as excellent bio-control agents against HS fungus. Thus, keeping these facts in mind, the present study was undertaken to profile the diversity of endophytic strains of *Bacillus* species from wheat genotypes and explore their antifungal activity for the field management of HS in wheat. 

A series of published literature indicated that investigation of the diversity of the plant microbiome is one of the potential approaches to identify novel and effective microorganisms as a fungal antagonist and plant growth promoter [56,57,58]. By following a cultivation-dependent approach, a total of 112 *Bacillus* strains were recovered from wheat ear heads of five different cultivars. These results are in harmony with earlier published literature, where a culture-dependent approach was employed to isolate the bacterial endophytes from different regions of the plant [59,60,61]. Interestingly, this study showed the endophytic association of *B. aerophilus*, *B. albus*, *B. atrophaeus*, *B. australimaris*, *B. badius*, *B. cabrialesii*, *B. cereus*, *B. clarus*, *B. coahuilensis*, *B. dafuensis*, *B. ferrooxidans*, *B. fungorum*, *B. glycinifermentans*, *B. haikouensis*, *B. halotolerans*, *B. haynesii*, *B. mojavensis*, *B. mycoides*, *B. nakamurai*, *B. paralicheniformis*, *B. paramycoides*, *B. pseudomycoides*, *B. rugosus*, *B. safensis*, *B. siamensis*, *B. stratosphericus*, *B. swezeyi*, *B. tequilensis*, *B. velezensis*, *B. wiedmannii*, *B. yapensis*, *B. zanthoxyli*, and *B. zhangzhouensis* with wheat ear head for the first time.

A large body of published reports on bacterial endophytes has focused on the plausible applications of isolated strains in managing agriculturally important diseases and established dual-culture assays as one of the golden standards for assessing the antagonistic effectiveness of isolated *Bacillus* strains prior to field validations [62,63]. Similarly, in the current study, a dual-culture growth inhibition test was performed, and the obtained results revealed 12.5% of the isolated *Bacillus* strains of antagonistic nature against *F. graminearum*. Based on the 16S rRNA sequence analysis performed in the present study, the antagonistic bacteria recovered from wheat ear heads were identified as *B. clarus* NOK09, *B. mojavensis* NOK16, *B. subtilis* NOK33, *B. rugosus* NOK47, *B. mojavensis* NOK52, *B. clarus* NOK59, *B. coahuilensis* NOK72, *Bacillus cabrialesii* NOK78, *B. cabrialesii* NOK82, *B. rugosus* NOK85, *B. amyloliquefaciens* NOK89, *B. australimaris* NOK95, *B. pumilus* NOK103, and *B. amyloliquefaciens* NOK109. These research findings agree with earlier reports, where *B. amyloliquefaciens*, *B. pumilus* and *B. subtilis* strains have been described as potential antagonists [64,65,66].

In the current study, three endophytic *Bacillus* strains (*B. clarus* NOK09, *B. subtilis* NOK33, and *B. amyloliquefaciens* NOK109) showing maximum bio-control activity against *F. graminearum* reflected strong character for potassium solubilization and siderophore production. These results are in harmony with earlier reports, where the presence of endophytic bacterial strains showing strong attributes of potassium solubilization and siderophore production were observed [67,68,69]. Kushwaha et al. [15] also recorded potassium solubilizing and siderophore-producing traits in endophytic strains of *B. cereus*, *B. amyloliquefaciens,* and *B. subtilis* subsp. *subtilis* from pearl millet host and further suggesting their function as a growth promoter, nutrient mobilizer, and disease defender. Here, it is important to mention that siderophore is a small, low molecular weight (500-1000 Daltons) iron-chelating agent which binds to the available iron making it unavailable for the phytopathogens. It also supports the plant by making iron easily available for various biological processes operating inside the plant system. Numerous recent studies reported siderophore production as the most common trait in bacterial endophytes associated with plants [69,70,71,72]. Apart from plant growth promoting traits, another mechanism that bacterial endophytes employ to combat fungal plant pathogens is the synthesis of extracellular hydrolytic enzymes [73]. In the present study, the three above-mentioned endophytic antagonistic *Bacillus* strains were also found to be positive for the production of chitinase, protease, lipase, amylase, and pectinase enzymes. Additionally, it appears that these hydrolytic enzymes could be responsible for the remarkable inhibition of *F. graminearum*. Here, it is worth mentioning that with the help of these extracellular hydrolytic enzymes, endophytes can penetrate the plant tissue and play a vital role in endophytic colonization and establishment inside the host plant [74,75]. 

PCR detection of AMP genes indicated that three antagonistic strains showing maximum bio-control activity against *F. graminearum* had three diverse antibiotic biosynthesis genes (*itu*D, *bmy*C, and *srf*A) that are linked with the production of the antibiotics iturin, bacillomycin, and surfactin. The obtained results are in conformity with other workers that revealed the prevalence and linkages of numerous antifungal peptide genes in diverse types of *Bacillus* strains [15,24,76,77,78,79,80]. Based on the published reports, it can be inferred that *Bacillus* species contain diverse types of AMP genes that have a role in the biosynthesis of antibiotics with specific modes of action [78,81]. For instance, the *itu* gene is essential in enhancing fungal cell membrane permeability [23]. On the other hand, the function of the *srf* gene is linked with biofilm synthesis [82] and *bmy* associated with the degradation and alterations of the cell wall and cell membrane of fungal hypha [76,83].

A large number of bacterial endophytes from diverse types of plants have been identified and evaluated for bio-control activities for several decades to control fungal diseases in India [15,84,85]. However, the research investigation with respect to the application of bio-control agents has been restricted to the in vitro identification and characterization of microbes against wheat pathogens. No field experimentation-based research report is documented for the suppression of wheat head scabs in India. Keeping these facts in mind, attempts were made to validate the hypothesis that combinations of endophytic bacteria are highly effective and versatile than individual endophytic strains in the bio-control of HS disease in wheat. Field testing of the biocontrol potential of endophytic bacterial strains (*B*. *clarus* NOK09, *B. subtilis* NOK33, and *B. amyloliquefaciens* NOK109) reflecting >70% mycelial growth inhibition against *F. graminearum* under in-vitro conditions in single or in combination was performed to test the hypothesis. Here, it is pertinent to mention that microbial consortia function in a strong and structured network and offers additional protection for the host to survive better than individual microorganisms under adverse conditions [86,87]. For instance, a combination of *Paenibacillus* sp. strain B2 and *Arhtrobacter* spp. strain AA was found more effective than a single-strain inoculum for promoting resistance against *Septoria tritici* blotch disease under drought stress [88]. Similarly, three strain mix (*B. subtilis* B2, *B. thuringiensis* B10, and *Enterobacter cloacae* B16) also showed noteworthy bio-control effects against *Sclerotinia* stem rot disease in tomatoes [89]. On parallel lines, the application of two formulated biological control agents (*Bacillus subtilis* RC 218 and *Brevibacillus* sp. RC 263) under semi-controlled field conditions resulted in a significant reduction in the HS severity level in wheat [90]. These studies strengthen the notion of evaluating the effect of the combination of antagonistic *Bacillus* strains of endophytic origin along with individual antagonists. The results emanating from the present investigation clearly evidenced the effectiveness of the combination of three distinct strains (*B*. *clarus* NOK09 + *B. subtilis* NOK33 and *B. amyloliquefaciens* NOK109) followed by the consortia of two strains in suppressing HS infection in wheat. However, positive and noteworthy antagonistic effects were achieved with all the test strains when used singly or in combination under in-vivo bio-control experiments. The reason that a single strain may be less effective can be associated with poor competitiveness against native microflora and fluctuating environmental conditions [91]. Here, it becomes important to underline that the repetitive success of the consortia in both the years under field conditions also supports the phenomenon of strain compatibility. In fact, the field results of triple endophyte consortia in bio-control were significantly different and better than the individual strain. These observations were also harmonized with the data obtained from the analysis of both years that yield gain was more pronounced in triple entophyte consortia relative to two strain consortia or individual strain application. These results were corroborated with the findings of Muhae-Ud-Din et al. [92], who noticed wheat grain yield gain by 30.8% over control check when a combination of endophytic microbes (*Bacillus* sp. MN54 + *Trichoderma* sp. MN6) was applied. Besides this, the results obtained in the current study with respect to the bio-control efficacy agree with earlier published literature, where a significant level of disease protection has been documented by spraying microbial consortia [87,88]. In addition, another imperative observation noticed in the current study was the additive effect of the microbial consortia composed of *B*. *clarus* NOK09 + *B. subtilis* NOK33 and *B. amyloliquefaciens* NOK109 when compared with the antagonistic effect of individual strain treatments on HS disease containment. Similar types of observation regarding the effectiveness of endophytic bacterial consortia in controlling the purple blotch disease and enhancing the growth and yield of shallots have been demonstrated by Resti et al. [93]. In a parallel fashion, Sundaramoorthy et al. [94] also documented the significant effect of the combination of *P. fluorescens* Pf1 and *B. subtilis* (EPCO16 and EPC5) in controlling the *Fusarium* wilt incidence in chilli by 17–30%. However, in the current study, better percent disease suppression in terms of HS incidence (62.52 to 63.86%) and severity (59.14–61.18%) was obtained when the combination of *B*. *clarus* NOK09 + *B. subtilis* NOK33 and *B. amyloliquefaciens* NOK109 spray were inoculated on the wheat plants. 

In conclusion, it can be inferred that wheat genotypes at the anthesis stage harbor 28 distinct species of *Bacillus* of entophytic nature, where only ten species displayed antagonism towards *F. graminearum*. Based on field results, it can be proposed that the combination of three endophytic *Bacillus* mixes (*B. clarus* NOK09 + *B. subtilis* NOK33 + *B. amyloliquefaciens* NOK109) in place of a single strain could be an important approach for the fieldmanagement of HS disease of wheat in an eco-friendly and sustainable manner.

## Figures and Tables

**Figure 1 pathogens-11-01088-f001:**
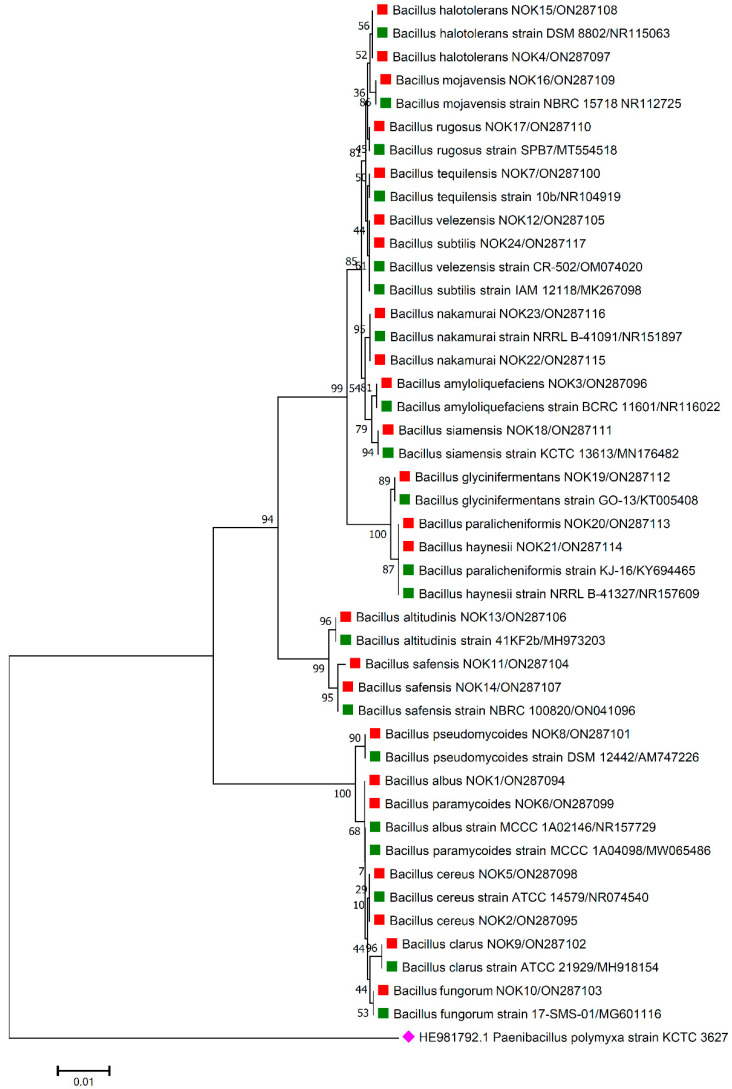
Phylogenetic tree based on partial 16S rRNA sequences of endophytic *Bacillus* strains (indicated by red colour square) allied with anthesis stage of wheat (cv. DBW187) along with those of maximum similar entries of type strains (indicated by green colour square) from database. The tree is constructed by neighbour-joining method with 1000 bootstrap replications. The scale bar represents the number of changes per base position. *Paenibacillus polymyxa* strain KCTC 3627 (HE981792.1) was used as an out-group strain (indicated by pink colour rhombus).

**Figure 2 pathogens-11-01088-f002:**
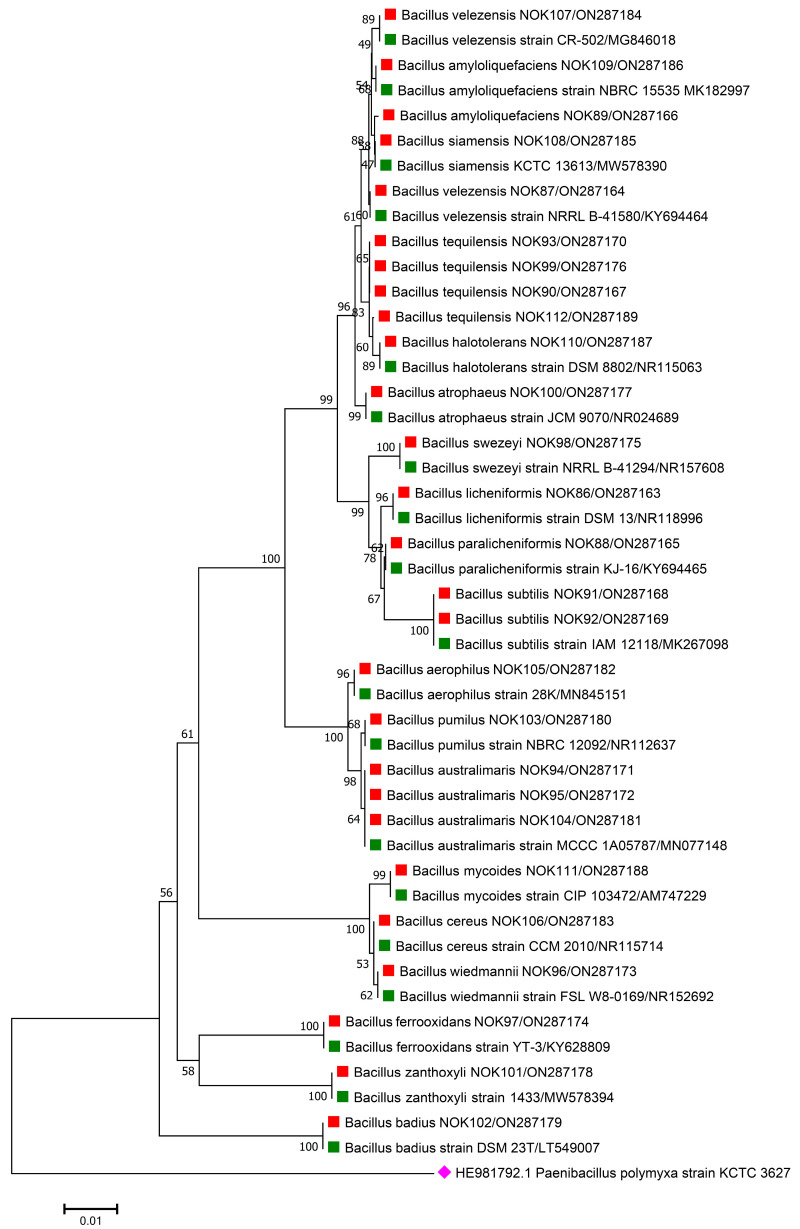
Phylogenetic tree based on partial 16S rRNA sequences of endophytic *Bacillus* strains (indicated by red colour squares) allied with anthesis stage of wheat (cv. HD2967) along with those of maximum similar entries of type strains (indicated by green colour squares) from database. The tree is constructed by neighbour-joining method with 1000 bootstrap replications. Scale bar represents the number of changes per base position. *Paenibacillus polymyxa* strain KCTC 3627 (HE981792.1) was used as an out-group strain (indicated by a pink colour rhombus).

**Figure 3 pathogens-11-01088-f003:**
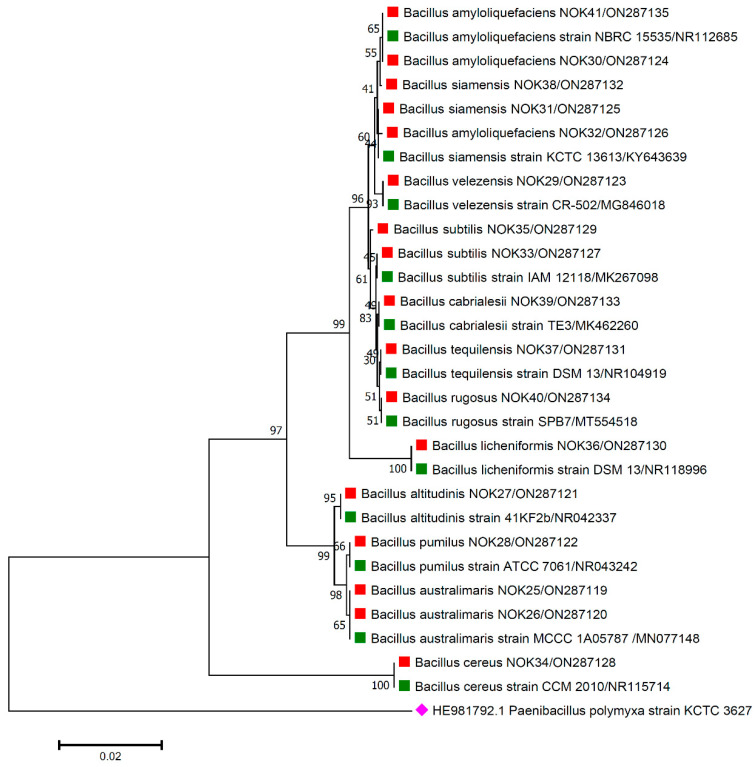
Phylogenetic tree based on partial 16S rRNA sequences of endophytic *Bacillus* strains (indicated by red colour squares) allied with anthesis stage of wheat (cv. PBW343) along with those of maximum similar entries of type strains (indicated by green colour squares) from database. The tree is constructed by neighbour-joining method with 1000 bootstrap replications. Scale bar represents the number of changes per base position. *Paenibacillus polymyxa* strain KCTC 3627 (HE981792.1) was used as an out-group strain (indicated by a pink colour rhombus).

**Figure 4 pathogens-11-01088-f004:**
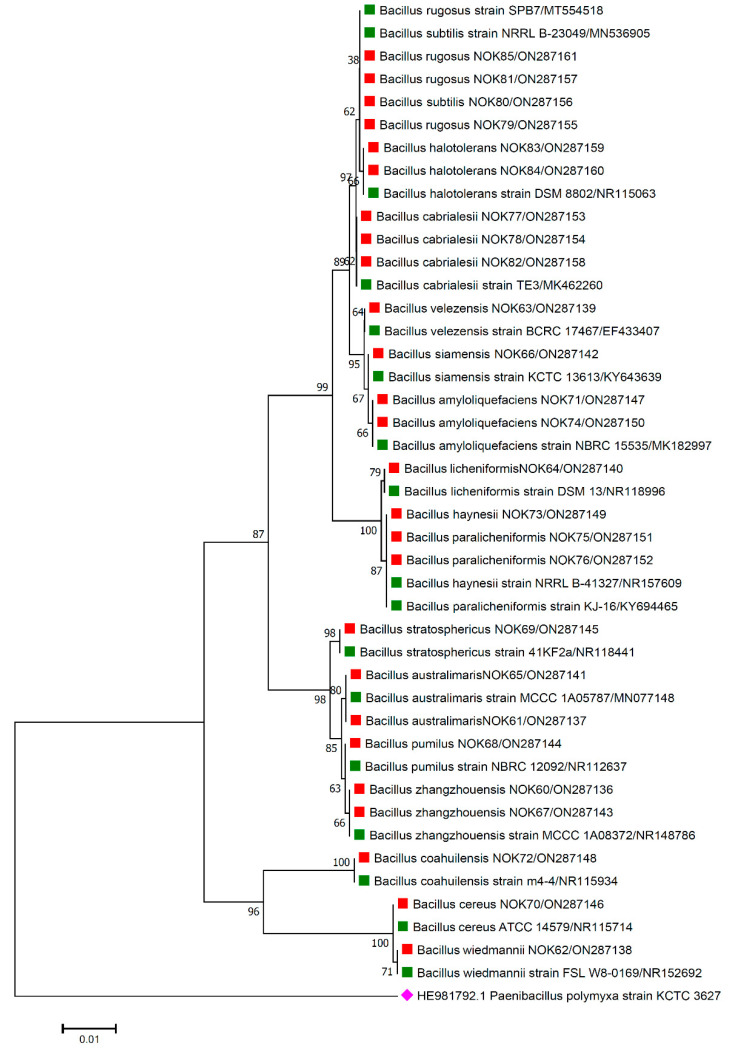
Phylogenetic tree based on partial 16S rRNA sequences of endophytic *Bacillus* strains (indicated by red colour squares) allied with anthesis stage of wheat (cv. HD3086) along with those of maximum similar entries of type strains (indicated by green colour squares) from database. The tree is constructed by neighbour-joining method with 1000 bootstrap replications. Scale bar represents the number of changes per base position. *Paenibacillus polymyxa* strain KCTC 3627 (HE981792.1) was used as an out-group strain (indicated by a pink colour rhombus).

**Figure 5 pathogens-11-01088-f005:**
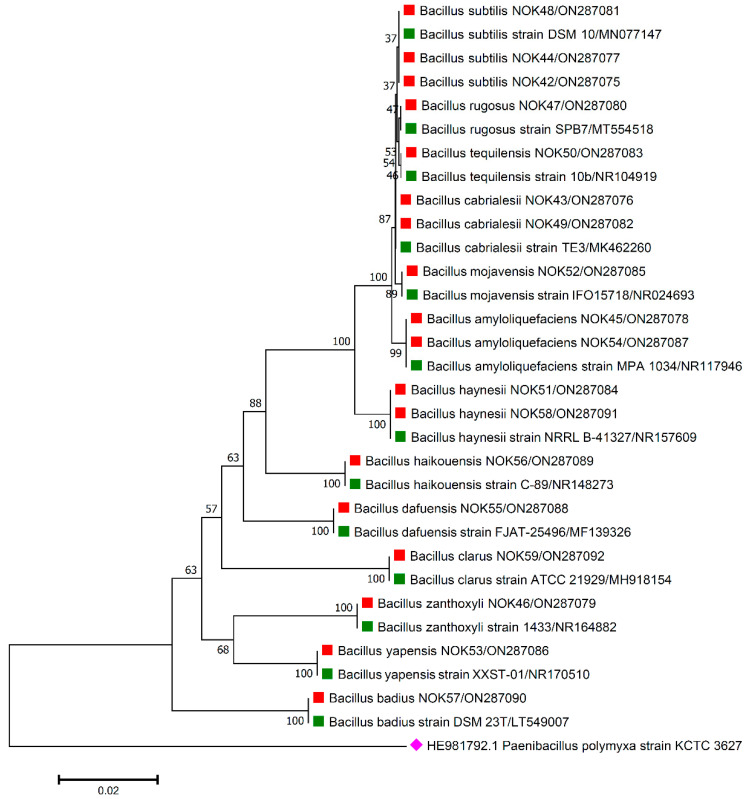
Phylogenetic tree based on partial 16S rRNA sequences of endophytic *Bacillus* strains (indicated by red square box symbols) allied with anthesis stage of wheat (cv. Agra Local) along with those of maximum similar entries of type strains (indicated by green square box symbols) from database. The tree is constructed by neighbour-joining method with 1000 bootstrap replications. Scale bar represents the number of changes per base position. *Paenibacillus polymyxa* strain KCTC 3627 (HE981792.1) was used as an out-group strain (indicated by a pink colour rhombus).

**Figure 6 pathogens-11-01088-f006:**
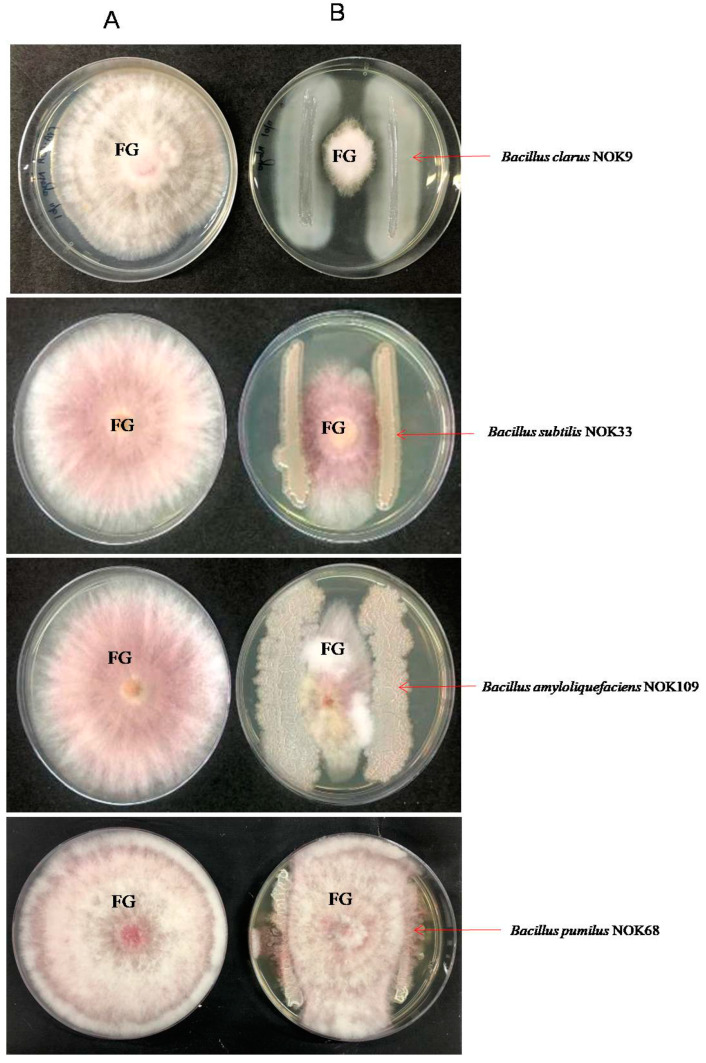
In-vitro interaction between endophytic *Bacillus* strains and *F. graminearum* NFG1 in a dual culture on NA: PDA plate at 5th day after incubation at 28 ± 2 °C. (**A**) A 5-mm agar plug of *F. graminearum* NFG1 on center of PDA plate and (**B**) Endophytic *Bacillus* strains inoculated on two corners of PDA plate with equal distance from the colony of *F. graminearum* placed at the centre. FG = *F.*
*graminearum* NFG1. *Bacillus pumilus* NOK68 serve as a neutral bacterial strain.

**Figure 7 pathogens-11-01088-f007:**
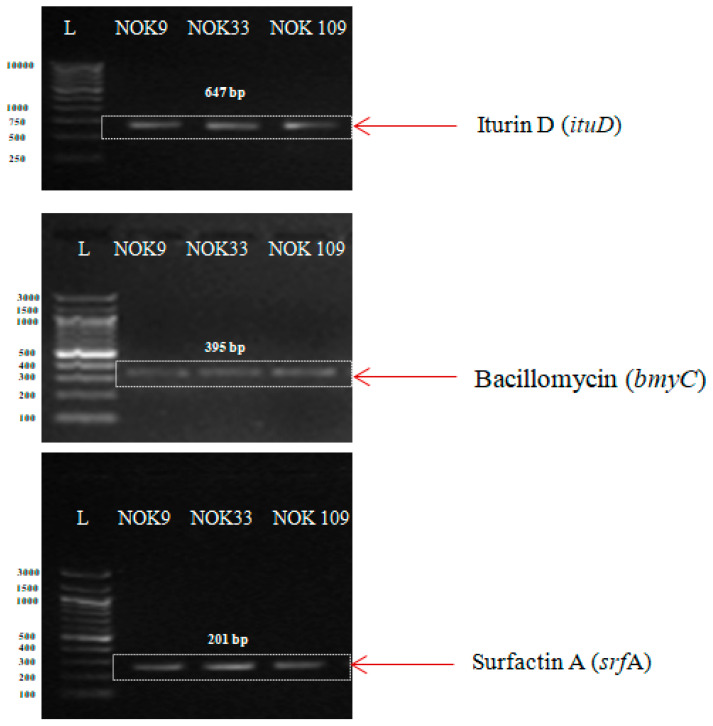
Agarose gel-electrophoresis of PCR products for revealing the presence of antimicrobial peptides (AMPs) i.e., iturin, bacillomycin, and surfactin A genes in promising antagonistic endophytic *Bacillus* strains. L is the DNA ladder. Numbers on left size indicates base pairs of the steps in the ladder.

**Figure 8 pathogens-11-01088-f008:**
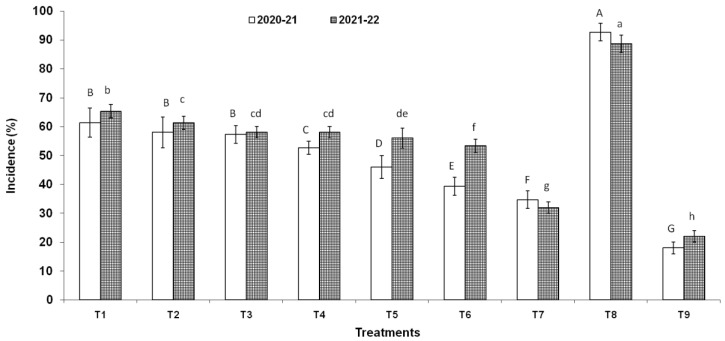
Disease incidence (%) of *Fusarium* head blight disease observed during 2020-21 and 2021-22 field trials conducted on susceptible wheat cultivar (cv. PBW343). On x axis, treatments are: T1 (NOK9 + NFG1), T2 (NOK33 + NFG1), T3 (NOK109 + NFG1), T4 (NOK9 + NBOK33 + NFG1), T5 (NOK 9 + NOK109 + NFG1), T6 (NOK33 + NOK109 + NFG1), T7 (NOK9 + NOK33 + NOK109 + NFG1), T8 (*F.*
*graminearum* NFG1), and T9 (Propiconazole @ 0.1%+ + NFG1). Data were analyzed for significance with an analysis of variance (ANOVA) followed by DMRT test (*p* = 0.05). Values with different letter indications represent a statistically significant difference.

**Figure 9 pathogens-11-01088-f009:**
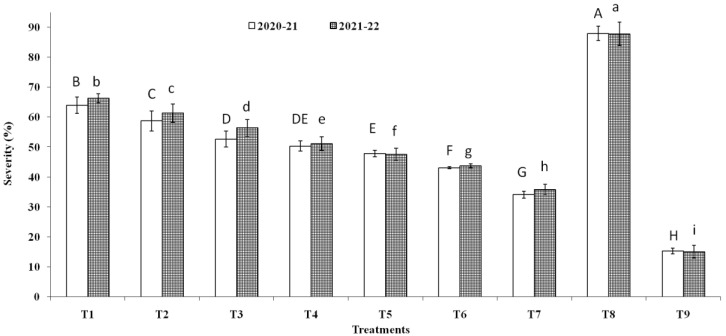
Disease severity (%) of *Fusarium* head blight disease observed during 2020-21 and 2021-22 field trials conducted on susceptible wheat cultivar (cv. PBW343). On X-axis, treatments are: T1 (NOK9 + NFG1), T2 (NOK33 + NFG1), T3 (NOK109 + NFG1), T4 (NOK9 + NBOK33 + NFG1), T5 (NOK 9 + NOK109 + NFG1), T6 (NOK33 + NOK109 + NFG1), T7 (NOK9 + NOK33 + NOK109 + NFG1), T8 (*F.*
*graminearum* NFG1), and T9 (Propiconazole @0.1% + NFG1). Data were analyzed for significance with analysis of variance (ANOVA) followed by DMRT test (*p* = 0.05). Values with different letter indications represent a statistically significant difference.

**Figure 10 pathogens-11-01088-f010:**
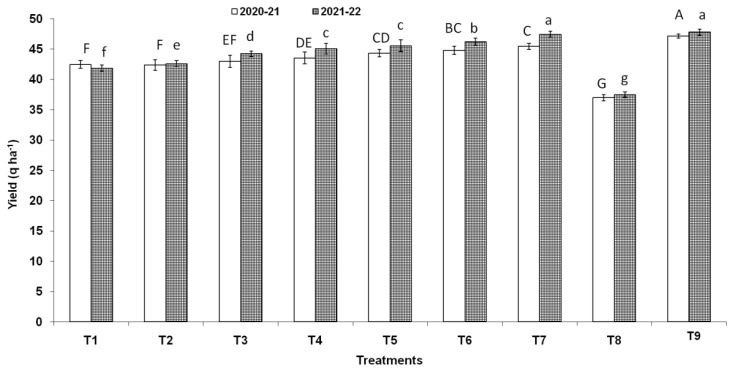
Influence of antagonists on the wheat (cv. PBW343) yield under field trials conducted during 2021–2021 and 2021–2022. On X-axis, treatments are: T1 (NOK9 + NFG1), T2 (NOK33 + NFG1), T3 (NOK109 + NFG1), T4 (NOK9 + NBOK33 + NFG1), T5 (NOK 9 + NOK109 + NFG1), T6 (NOK33 + NOK109 + NFG1), T7 (NOK9 + NOK33 + NOK109 + NFG1), T8 (*F.*
*graminearum* NFG1), and T9 (Propiconazole @ 0.1% + NFG1). Data were analyzed for significance with analysis of variance (ANOVA) followed by DMRT test (*p* = 0.05). Values with different letter indications represent a statistically significant difference Y-axis represent grain yield in q ha^−1^.

**Table 1 pathogens-11-01088-t001:** List of primers and thermal profile employed for PCR based amplification of target genomic region of endophytic *Bacillus* strains.

Target Region	Gene	Primer Sequences (5′–3′)	PCR Profile	Product Size (bp)	References
16S rRNA	27F	AGAGTTTGATCMTGGCTCAG	Initial denaturation at 95 °C for 4 min, followed by 35 cycles of denaturation, annealing, and elongation at 95 °C for 1 min, 56 °C for 1 min and 72 °C for 90 s, respectively. The final extension step was done at 72 °C for 10 min		[43]
	1525R	AAGGAGGTGWTCCARCC		~1500	
Surfactin	*srfA*-F	TCGGGACAGGAAGACATCAT CCACTCAAACGGATAATCCTGA	Initial denaturation at 95 °C for 4 min, 35 cycles of 95 °C for 35 s, 58 °C for 30 s, and 72 °C for 45 s, final extension 72 °C for 10 min	201	[44]
	*srf*A-R				
Bacillomycin	*bmy*C-F	TGAAACAAAGGCATATGCTC AAAAATGCATCTGCCGTTCC	Initial denaturation at 94 °C for 4 min, 40 cycles of 95 °C for 60 s, 56 °C for 30 s, and 72 °C for 45 s, final extension 72 °C for 10 min	395	[45]
	*bmy*C-R				
Iturin	*Itu*D-F	GATGCGATCTCCTTGGATGT ATCGTCATGTGCTGCTTGAG	Initial denaturation at 95 °C for 4 min, 35 cycles of 95 °C for 35 s, 60 °C for 30 s, and 72 °C for 45 s, final extension 72 °C for 10 min	647	[44]

**Table 2 pathogens-11-01088-t002:** In-vitro evaluation of the antagonism against *Fusarium graminearum* and different activities of production of siderophore and hydrolytic enzymes and potassium solubilisation displayed by endophytic *Bacillus* strains isolated at anthesis stage from different wheat genotypes.

Strain	Wheat Genotype	^#^ PDC (%)	Diameter of Clear Zone (mm)						
			Siderophore	K-SolubilIzation	Protease	Lipase	Amylase	Chitinase	Pectinase
NOK09	DBW187	*77.3 ± 0.14 ^b^	20.95 ± 1.01 ^a^	17.45 ± 1.84 ^a^	13.50 ± 0.55 ^a^	21.47 ± 1.71 ^a^	19.12 ± 1.12 ^a^	28.43 ± 0.65 ^a^	22.53 ± 1.01 ^a^
NOK16	DBW187	36.2 ± 0.62 ^h^	10.26 ± 0.14 ^e^	10.34 ± 2.43 ^b^	10.08 ± 0.81 ^c^	-	13.42 ± 1.02 ^b^	20.12 ± 1.32 ^b^	18.03 ± 1.05 ^b^
NOK33	PBW343	71.9 ± 0.23 ^c^	20.62 ± 1.41 ^a^	16.58 ± 1.41 ^a^	13.14 ± 2.20 ^a^	20.10 ± 1.04 ^a^	18.62 ± 1.21 ^a^	28.22 ± 1.08 ^a^	22.42 ± 1.20 ^a^
NOK47	Agra Local	69.04 ± 0.42 ^d^	13.24 ± 1.43 ^bc^	9.42 ± 1.31 ^bc^	10.42 ± 0.55 ^c^	13.71 ± 1.43 ^c^	12.02 ± 1.01 ^bc^	10.12 ± 1.19 ^f^	13.62 ± 1.08 ^e^
NOK52	Agra Local	66.66 ± 0.52 ^e^	11.01 ± 1.12 ^d^	10.85 ± 1.08 ^b^	10.21 ± 1.01 ^c^	11.07 ± 1.76 ^cd^	11.32 ± 1.72 ^cd^	14.30 ± 1.33 ^d^	15.74 ± 1.09 ^cd^
NOK59	Agra Local	69.04 ± 0.52 ^d^	10.11 ± 1.08 ^d^	10.86 ± 0.95 ^b^	10.41 ± 0.64 ^c^	12.27 ± 1.08 ^c^	13.12 ± 1.07 ^b^	11.40 ± 1.07 ^f^	13.12 ± 1.02 ^e^
NOK68	HD3086	**ND	ND	ND	ND	ND	ND	ND	ND
NOK72	HD3086	53.4 ± 0.42 ^g^	14.32 ± 0.88 ^b^	-	10.70 ± 0.52 ^c^	-	11.02 ± 1.0 ^cd^	16.46 ± 0.26 ^d^	12.20 ± 1.12 ^ef^
NOK78	HD3086	61.9 ± 0.56 ^f^	10.51 ± 1.75 ^de^	8.95 ± 0.98 ^c^	9.14 ± 2.08 ^cd^	12.54 ± 1.23 ^c^	10.32 ± 1.06 ^d^	18.73 ± 0.35 ^c^	11.85 ± 1.21 ^ef^
NOK82	HD3086	69.04 ± 0.52 ^d^	12.35 ± 2.01^c^	10.45 ± 1.84 ^b^	10.20 ± 0.81 ^c^	-	13.25 ± 1.02 ^b^	13.43 ± 0.65 ^e^	11.04 ± 1.01 ^ef^
NOK85	HD3086	64.28 ± 0.45 ^e^	14.35 ± 2.01 ^b^	-	11.08 ± 0.92 ^bcd^	15.08 ± 1.27 ^bc^	12.42 ± 1.52 ^c^	18.12 ± 1.42 ^c^	14.04 ± 1.15 ^cd^
NOK89	HD2967	61.9 ± 0.58 ^f^	10.62 ± 2.31 ^de^	11.08 ± 1.41 ^b^	10.14 ± 2.05 ^c^	12.10 ± 1.04 ^c^	11.22 ± 2.21 ^cd^	13.42 ± 1.18 ^e^	16.42 ± 1.22 ^c^
NOK95	HD2967	36.2 ± 0.59 ^h^	13.24 ± 1.20 ^bc^	-	12.42 ± 0.95 ^b^	10.71 ± 1.43 ^d^	12.02 ± 1.07 ^bc^	13.12 ± 1.09 ^e^	14.62 ± 1.18 ^cd^
NOK103	HD2967	60.3 ± 0.82 ^fg^	12.01 ± 1.40 ^bcd^	11.85 ± 1.90 ^b^	11.21 ± 1.31 ^bcd^	11.07 ± 1.46 ^cd^	10.52 ± 1.52 ^de^	14.30 ± 1.43 ^e^	15.74 ± 1.09 ^cd^
NOK109	HD2967	79.4 ± 0.52 ^a^	21.11 ± 1.08 ^a^	17.86 ± 0.95 ^a^	14.21 ± 1.11 ^a^	22.07 ± 1.26 ^a^	19.32 ± 1.72 ^a^	29.30 ± 1.44 ^a^	23.74 ± 1.29 ^a^

Data were analyzed for significance with analysis of variance (ANOVA) followed by DMRT (*p* = 0.05). * Values with different alphabets indicate statistically significant difference. Data shown correspond to mean of three replications ± the standard deviation; ^#^ PDC= Plant disease over control; ** ND = Not determined.

**Table 3 pathogens-11-01088-t003:** Effect of endophytic *Bacillus* strains on *Fusarium* head scab incited of wheat (cv. PBW343).

Treatment(s)	*Disease Index (%)	
	2020–2021	2021–2022
T1 (NOK9 + NFG1)	39.19	43.28
T2 (NOK33 + NFG1)	34.05	37.62
T3 (NOK109 + NFG1)	30.19	32.70
T4 (NOK9 + NOK33 + NFG1)	26.50	29.68
T5 (NOK9 + NOK109 + NFG1)	21.99	26.63
T6 (NOK33+ NOK109 + NFG1)	16.94	23.33
T7 (NOK9 +NOK33 + NOK109 + NFG1)	11.82	11.46
T8 (NFG1 only)	81.47	77.85
T9 (Propiconazole @ 0.1% + NFG1)	2.75	3.31

*Disease index (%) was calculated by using the formula: (incidence x severity)/100). Fusarium head scab (HS) disease severity (%) of symptomatic spikelets recorded 15 days post inoculation at anthesis stage.

## Data Availability

The data are contained within the article or gene sequences generated in this study were deposited in GenBank under the accession numbers mentioned in Figure 1, Figure 2, Figure 3, Figure 4 and Figure 5.

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
