# Peer review of "Diversity and Exploration of Endophytic Bacilli for the Management of Head Scab (Fusarium graminearum) of Wheat"

_pathogens, 2022, doi:10.3390/pathogens11101088_

Round 1

Reviewer 1 Report

In this article, the authors isolated a large number of endophytic bacilli from five different wheat genotypes and identified 14 strains that displayed an antagonistic effect against Fusarium graminearum. In two year consecutive field study, three strain consortia could effectively control head scab disease index. Biosynthesis genes of three antibiotics were detected in these strains. Overall, this paper was well-written and the conclusion was supported by the experiments. Some points need to be addressed before publication:

1.    Figures 1 to 5, some figures are not in high resolution and the colored symbols before strains should be explained in the caption.

2.       About the field experiment, this manuscript use only tables (8-11) to demonstrate the control effect, can the authors provide some figures to support this?

3.       Figure 7, the DNA ladder should be labeled with standards.

4.       Grammar and spelling mistakes:

Line 21, endophyttic

Line 62, influence

Line 86, has the reported

Line 229, deign

Line 366, chose

Line 435, found

Author Response

Comments: In this article, the authors isolated a large number of endophytic bacilli from five different wheat genotypes and identified 14 strains that displayed an antagonistic effect against Fusarium graminearum. In two year consecutive field study, three strain consortia could effectively control head scab disease index. Biosynthesis genes of three antibiotics were detected in these strains. Overall, this paper was well-written and the conclusion was supported by the experiments. Some points need to be addressed before publication:

Response: Thanks for the critical and valuable comments. The manuscript has been thoroughly revised in light of points suggested by the esteemed reviewer. All the queries have been attended and changes were highlighted in yellow in the revised version of the manuscript.

Comments: Figures 1 to 5, some figures are not in high resolution and the colored symbols before strains should be explained in the caption.

Response: The Figures 1 to 5 have been redrawn in high resolution and inserted in the revised version of manuscript. The information pertaining to the colored symbols before strains have also been provided in all the figure captions and highlighted in yellow in the submitted manuscript.

Comment: About the field experiment, this manuscript use only tables (8-11) to demonstrate the control effect, can the authors provide some figures to support this?

Response: The data related to field experiments has been provided in the form Figure 8- Fig 11.

Comment: Figure 7, the DNA ladder should be labeled with standards.

Response: Desired information has been inserted in the revised version of manuscript.

Comment:  Grammar and spelling mistakes:

Line 21, endophyttic

Line 62, influence

Line 86, has the reported

Line 229, deign

Line 366, chose

Line 435, found

Response: The manuscript has been thoroughly checked for Grammar and spelling mistakes with the help of native English speaker.

Reviewer 2 Report

The study from Kaul and coauthors describes the isolation and characterization of endophytic bacterial strains from the Bacillus genus for their biocontrol potential against Fusarium Head Blight either alone or in combination.

The manuscript is rather difficult to follow and English needs strong improvement. Experiments are overall well conducted with appropriate controls apart from the antagonism assays (see below). Data analyses are appropriate except from the phylogeny (see below).

Phylogenetic analyses should be improved. Trees are of poor quality which is likely due to the choice of a too distant bacterial species as outgroup. The authors may rather use sequences from bacterial species closer to the Bacillus genus (Paenibacillus, Brevibacillus, …) which will undoubtedly give rise to better resoluted phylogenetic trees. The description of the analysis method should be improved in the Materials and methods.

Concerning the morphological characterization of the isolated Bacillus strains (not only biochemical tests, as mentioned by the authors line 131), the authors poorly comment on the fact that the best antagonistic strains are also the ones showing the best results for siderophore production as well as hydrolytic enzyme activities. This should be added.

Antagonistic tests lack controls, for example using a more neutral bacterial strain than the Bacillus strains tested.

Field assays are well conducted with appropriate experimental design and analyses are appropriate.

The discussion is poor and do not refer at all to the characteristics of the three bacterial strains.

In conclusion, even if of global interest in the context of an environment-friendly agriculture, a number of improvements should be brought both to analyses and to the manuscript to deserve publication in Pathogens.

Author Response

Comments and Suggestions for Authors: The study from Kaul and coauthors describes the isolation and characterization of endophytic bacterial strains from the Bacillus genus for their biocontrol potential against Fusarium Head Blight either alone or in combination. The manuscript is rather difficult to follow and English needs strong improvement. Experiments are overall well conducted with appropriate controls apart from the antagonism assays (see below). Data analyses are appropriate except from the phylogeny (see below).

Response: Thanks for the critical and valuable comments. As per the suggestion, the manuscript has been thoroughly checked and revised for grammar and spelling mistakes with the help of native English speaker.

Comment: Phylogenetic analyses should be improved. Trees are of poor quality which is likely due to the choice of a too distant bacterial species as outgroup. The authors may rather use sequences from bacterial species closer to the Bacillus genus (Paenibacillus, Brevibacillus, …) which will undoubtedly give rise to better resoluted phylogenetic trees. The description of the analysis method should be improved in the Materials and methods.

Response: Desired information pertaining to the Phylogenetic analyses has been provided both in the text as figure caption ijn the revised versiuon of manuscript. The phylogenetic analysis has been improved by considering Paenibacillus polymyxa strain KCTC 3627 (HE981792.1) as an out-group strain.

Comment: Concerning the morphological characterization of the isolated Bacillus strains (not only biochemical tests, as mentioned by the authors line 131), the authors poorly comment on the fact that the best antagonistic strains are also the ones showing the best results for siderophore production as well as hydrolytic enzyme activities. This should be added.

Response: Desired information related to the results of siderophore production as well as hydrolytic enzyme activities has been appended in the manuscript and highlight in yellow in the revised version of manuscript.

Comment: Antagonistic tests lack controls, for example using a more neutral bacterial strain than the Bacillus strains tested.

Response: Bacillus pumilus NOK68 was taken as a neutral bacterial strain during antagonistic test as additional control check as per the reviewers’ suggestion. The new image (Fig 6) of same has been inserted in the revised version of manuscript.

Comment: Field assays are well conducted with appropriate experimental design and analyses are appropriate.

Response: Thanks for the critical and valuable comments along with encouraging words.

Comment: The discussion is poor and do not refer at all to the characteristics of the three bacterial strains.

Response: The desired information has been included in the discussion section and highlighted in yellow.

Comment: In conclusion, even if of global interest in the context of an environment-friendly agriculture, a number of improvements should be brought both to analyses and to the manuscript to deserve publication in Pathogens.

Response: All the points mentioned for the improvements of the manuscript have been included in revised version of the manuscript.

Reviewer 3 Report

Dear Authors,

in my opinion prepared by you manuscript is very interesting both in practical, as well in cognitive context and contributes a lot to phytopathology, mycology, plant protection and ecology. The research conducted by you will certainly draw the attention of the scientific community on the possible alternative strategies of plant protection and especially wheat. In summary, this work undoubtedly has a strong applicative potential and obtained by you results should be patented.

All the figures and tables are appropriate for this type of article. In general, the paper has a logical flow. The abstract well correspond with the main aspects of the work. Nevertheless, as a reviewer I am obligated to pay attention even to less important weak points of this work and all mentioned below comments should be carefully considered.

Line 4

With the surname of the second corresponding author, the asterisk is smaller or in a different format. Please standardize the spelling.

Line 6

With the affiliation of the author of Deepti Singh, the name of the country was not indicated

Line 9 and 14

To the best of my knowledge should be ,,Fusarium graminearum”. Please check and correct where necessary throughout the manuscript.

Line 15

As I know should be ,,…was confirmed by matching 16S RNA sequences with type strains…”

Line 21

I would like to suggest small correction, namely ,,Furthermore, the three endophytic Bacillus strains showing the strongest antagonistic effect …”

Line 22

In my opinion ,, … > 70% of growth inhibition of fungal mycelium …” sounds better

Line 22

In my opinion ,, … assessed (or evaluated) by antagonistic activity test were selected for field experiments …” sounds better and correct.

Lines 25-26

,, … three genes involved in antibiotic biosynthesis pathway …” sounds really better

Line 27

As I know should be ,,Additional attributes such as …”

Line 27

To the best of my knowledge should be ,,solubilization"instead of ,,solubilizion"

Line 44

In my opinion should be ,,Published literature revealed that crop rotation, … “

Line 49

,, … due to lack of …” sounds better

Line 56

In my opinion ,, … mutations resulting in resistance phenotypes …” sounds correct

Line 58

I would suggest including the name of this particular fungicide

Line 62

I think there should be ,, … is highly influenced by …”

Line 86

As I suspect should be ,,… has been reported to enhance…”

Line 87

There is something wrong ,, Pan et al. (17 2015)”

Line 88

Should be ,,endophytic” instead of ,,endopyhtic”

Line 100

There is something wrong ,, … as the highly susceptibility window…”

Line 104

In my opinion sounds better ,, Thus, the present study …”

Line 128

To the best of my knowledge should be ,,Luria Bertani agar” and ,,tryptic soy agar”

Line 131

There are two commas

Line 130

The "NA Petri plates" abbreviation should be developed on first use

Line 138

There is something wrong ,,Amplification of amplification of 16S rDNA region…”

Lines 139-140

,,…and conditions of PCR reaction are described in Table 1…” sounds better

Line 151, Table 1

As I know should be ,,bacillomycin”

Line 152, Figure 1

I would like to recommend improving the readability of Figure 1

Line 153

In my opinion ,,Phylogenetic tree based on partial 16S rDNA sequences” sounds better

Line 155

As I know should be ,,1000 bootstrap replications”. The same recommendation applies to figures 2-5.

Line 179

As I know should be ,, 5 mm diameter segment”

Line 186

Please, specify whether these were three independent repetitions?

Line 214

As I know should be ,,cellulase”

Line 220

Please, specify whether these were three independent repetitions?

Line 225

According to Table 1 should be ,,Surfactin, Bacillomycin and Iturin genes”

Line 229

As I assume should be complete randomized block design (CARD) designed with three replicates.

Line 230

There is something wrong

Line 234

In my opinion correction is necessary. In my opinion ,,Species diversity of endophytic Bacillus spp. in tissues of wheat of different genotypes” sounds correct and more appropriate.

Line 263, Table 2

There should be ,,Fusarium graminearum” instead of ,,Fusarium gramianreum”

Line 269

,,Strain characterization for potassium solubilization, siderophores release and hydrolytic enzyme activity” sounds better and correct.

Line 282

In my opinion should be ,,antagonistic strains” instead of ,,antagonists strains”

Line 283

Should be ,,endophytic”

Line 290

Instead of ,,in consortium"I would like to recommend using ,,in combination". Please check and correct where necessary throughout the manuscript.

Line 294

,,sprayed with a combination of three strains composed …” sounds better and correct

Line 308

In my opinion small inhibition zone is visible only in case of Bacillus clarus. Maybe Authors have better photos that clearly show a typical growth inhibition zone?

Line 342

I suspect that there should be ,,Disease index (%) was calculated by using the formula…” instead of ,, Disease index and was calculated by using the formula”

Line 357

As I know should be ,,Furthermore, the farmers many times”

Line 377

There should be ,,endophytes” instead of ,,entophytes”

Line 407

Instead of ,,strains have been acknowledged as potential antagonists” I would like to recommend ,, strains have been described as potential antagonists”

Line 426

Instead of ,,consortia of endophytic bacteria” I would like to recommend ,,combinations of endophytic bacteria”

Lines 476-477

Instead of ,,…it can be proposed that consortium of three endophytic Bacillus mix…” I would like to recommend ,,it can be proposed that combination of three endophytic Bacillus mix”

Author Response

Comment: In my opinion prepared by you manuscript is very interesting both in practical, as well in cognitive context and contributes a lot to phytopathology, mycology, plant protection and ecology. The research conducted by you will certainly draw the attention of the scientific community on the possible alternative strategies of plant protection and especially wheat. In summary, this work undoubtedly has a strong applicative potential and obtained by you results should be patented.

Response: Thanks for the critical, valuable and encouraging comments for the improvements of the manuscript and quality presentation of research data.

Comment: All the figures and tables are appropriate for this type of article. In general, the paper has a logical flow. The abstract well corresponds with the main aspects of the work. Nevertheless, as a reviewer I am obligated to pay attention even to less important weak points of this work and all mentioned below comments should be carefully considered.

Line 4

With the surname of the second corresponding author, the asterisk is smaller or in a different format. Please standardize the spelling.

Line 6

With the affiliation of the author of Deepti Singh, the name of the country was not indicated

Line 9 and 14

To the best of my knowledge should be ,,Fusarium graminearum”. Please check and correct where necessary throughout the manuscript.

Line 15

As I know should be ,,…was confirmed by matching 16S RNA sequences with type strains…”

Line 21

I would like to suggest small correction, namely ,,Furthermore, the three endophytic Bacillus strains showing the strongest antagonistic effect …”

Line 22

In my opinion ,, … > 70% of growth inhibition of fungal mycelium …” sounds better

Line 22

In my opinion ,, … assessed (or evaluated) by antagonistic activity test were selected for field experiments …” sounds better and correct.

Lines 25-26

,, … three genes involved in antibiotic biosynthesis pathway …” sounds really better

Line 27

As I know should be ,,Additional attributes such as …”

Line 27

To the best of my knowledge should be ,,solubilization"instead of ,,solubilizion"

Line 44

In my opinion should be ,,Published literature revealed that crop rotation, … “

Line 49

,, … due to lack of …” sounds better

Line 56

In my opinion ,, … mutations resulting in resistance phenotypes …” sounds correct

Line 58

I would suggest including the name of this particular fungicide

Line 62

I think there should be ,, … is highly influenced by …”

Line 86

As I suspect should be ,,… has been reported to enhance…”

Line 87

There is something wrong ,, Pan et al. (17 2015)”

Line 88

Should be ,,endophytic” instead of ,,endopyhtic”

Line 100

There is something wrong ,, … as the highly susceptibility window…”

Line 104

In my opinion sounds better ,, Thus, the present study …”

Line 128

To the best of my knowledge should be ,,Luria Bertani agar” and ,,tryptic soy agar”

Line 131

There are two commas

Line 130

The "NA Petri plates" abbreviation should be developed on first use

Line 138

There is something wrong ,,Amplification of amplification of 16S rDNA region…”

Lines 139-140

,,…and conditions of PCR reaction are described in Table 1…” sounds better

Line 151, Table 1

As I know should be ,,bacillomycin”

Line 152, Figure 1

I would like to recommend improving the readability of Figure 1

Line 153

In my opinion ,,Phylogenetic tree based on partial 16S rDNA sequences” sounds better

Line 155

As I know should be ,,1000 bootstrap replications”. The same recommendation applies to figures 2-5.

Line 179

As I know should be ,, 5 mm diameter segment”

Line 186

Please, specify whether these were three independent repetitions?

Line 214

As I know should be ,,cellulase”

Line 220

Please, specify whether these were three independent repetitions?

Line 225

According to Table 1 should be ,,Surfactin, Bacillomycin and Iturin genes”

Line 229

As I assume should be complete randomized block design (CARD) designed with three replicates.

Line 230

There is something wrong

Line 234

In my opinion correction is necessary. In my opinion ,,Species diversity of endophytic Bacillus spp. in tissues of wheat of different genotypes” sounds correct and more appropriate.

Line 263, Table 2

There should be ,,Fusarium graminearum” instead of ,,Fusarium gramianreum”

Line 269

,,Strain characterization for potassium solubilization, siderophores release and hydrolytic enzyme activity” sounds better and correct.

Line 282

In my opinion should be ,,antagonistic strains” instead of ,,antagonists strains”

Line 283

Should be ,,endophytic”

Line 290

Instead of ,,in consortium"I would like to recommend using ,,in combination". Please check and correct where necessary throughout the manuscript.

Line 294

,,sprayed with a combination of three strains composed …” sounds better and correct

Line 308

In my opinion small inhibition zone is visible only in case of Bacillus clarus. Maybe Authors have better photos that clearly show a typical growth inhibition zone?

Line 342

I suspect that there should be ,,Disease index (%) was calculated by using the formula…” instead of ,, Disease index and was calculated by using the formula”

Line 357

As I know should be ,,Furthermore, the farmers many times”

Line 377

There should be ,,endophytes” instead of ,,entophytes”

Line 407

Instead of ,,strains have been acknowledged as potential antagonists” I would like to recommend ,, strains have been described as potential antagonists”

Line 426

Instead of ,,consortia of endophytic bacteria” I would like to recommend ,,combinations of endophytic bacteria”

Lines 476-477

Instead of ,,…it can be proposed that consortium of three endophytic Bacillus mix…” I would like to recommend ,,it can be proposed that combination of three endophytic Bacillus mix”

Response: All the points mentioned above has been incorporated carefully and highlighted in yellow in the revised version of the manuscript. The revised manuscript has been thoroughly checked with the help of native English speaker for any typos or grammatical errors too.

Round 2

Reviewer 2 Report

Review on revised manuscript Pathogens-1910884

In the revised version of manuscript 1910884, the authors have significantly improved a number of items that had been pointed out in the initial review.

Phylogenetic analyses have been greatly improved by changing the bacterial species used as outgroup. Moreover, the authors have added important information both in the materials and methods section and in Figures 1-5 captions.

The authors have also included a “neutral” Bacillus strain (NOK68) obviously exhibiting less antagonistic activity as shown in Figure 6. The authors should add the percentage of Fusarium graminearum growth inhibition at least in the paragraph describing the results of these experiments.

A number of mistakes of Table or Figure numbers or captions are remaining:

- lines 281 and 305: change Table 1 into Table 2

- line 314: Fig 6 and Fig 7 should be changed to Fig 9

- lines 345 and 357: 2021-21 should be changed into 2020-21

The paragraph concerning the detection of AMP genes (lines 374-379) could be moved before the one dealing with filed assays, this would be more logical.

Finally, the manuscript still requires strong improvement of English language (see some comments in the attached file).

Author Response

Comments: In the revised version of manuscript 1910884, the authors have significantly improved a number of items that had been pointed out in the initial review. Phylogenetic analyses have been greatly improved by changing the bacterial species used as outgroup. Moreover, the authors have added important information both in the materials and methods section and in Figures 1-5 captions.

The authors have also included a “neutral” Bacillus strain (NOK68) obviously exhibiting less antagonistic activity as shown in Figure 6. The authors should add the percentage of Fusarium graminearum growth inhibition at least in the paragraph describing the results of these experiments.

Response: Thanks for the encouraging and constructive remarks of the expert. The desired information pertaining to the percentage of Fusarium graminearum growth inhibition in paragraph describing the results of the antagonistic experiments have been appended and highlighted in yellow in the submitted version of the manuscript.

Comments: A number of mistakes of Table or Figure numbers or captions are remaining:

- lines 281 and 305: change Table 1 into Table 2

- line 314: Fig 6 and Fig 7 should be changed to Fig 9

- lines 345 and 357: 2021-21 should be changed into 2020-21

The paragraph concerning the detection of AMP genes (lines 374-379) could be moved before the one dealing with filed assays, this would be more logical.

Response: All the suggestions and above mentioned recommendations made by the expert has been incorporated at aproparite palce in the revised version of manuscript and highlighted in yellow. The Tables number, Figure numbers and captions are thoroughly checked and corrected where ever erroneous and highlighted in yellow.

Comment: Finally, the manuscript still requires strong improvement of English language (see some comments in the attached file).
Response: The manuscript has been thoroughly checked for typos and grammar errors and all the correction. All the comments made by the expert in eth attached file have been attended in the revised version of the manuscript and highlighted in yellow.